# Genome-Wide Association Study of Body Conformation Traits by Whole Genome Sequencing in Dazu Black Goats

**DOI:** 10.3390/ani12050548

**Published:** 2022-02-23

**Authors:** Bowen Gu, Ruifan Sun, Xingqiang Fang, Jipan Zhang, Zhongquan Zhao, Deli Huang, Yuanping Zhao, Yongju Zhao

**Affiliations:** 1College of Animal Science and Technology, Southwest University, Chongqing 400715, China; gubw04@163.com (B.G.); sunruifan111@163.com (R.S.); 15881044561@163.com (X.F.); jpanzhang@live.com (J.Z.); zhaozhongquan@swu.edu.cn (Z.Z.); 2Chongqing Key Laboratory of Herbivore Science, Chongqing 400715, China; 3Chongqing Engineering Research Center for Herbivores Resource Protection and Utilization, Chongqing 400715, China; 4Tengda Animal Husbandry Co., Ltd., Chongqing 402360, China; huangdl116@163.com; 5Dazu County Agriculture and Rural Committee, Chongqing 402360, China; dzxmz@163.com

**Keywords:** body conformation traits, whole genome sequencing, genome-wide association study, single nucleotide polymorphism, goat

## Abstract

**Simple Summary:**

Body conformation traits are economically important in the goat meat industry. Good growth performance in goats, including an accelerated growth rate, can improve carcass weight and meat yield. The identification of genetic variants associated with these traits provides a basis for the genetic improvement of growth performance. In this study, we measured six body conformation traits, including body height, body length, cannon circumference, chest depth, chest width, and heart girth. By a genome-wide association study of a Chinese meat goat breed, 53 significant single nucleotide polymorphisms and 42 candidate genes associated with these traits were detected. These findings improve our understanding of the genetic basis of body conformation traits in goats.

**Abstract:**

Identifying associations between genetic markers and economic traits has practical benefits for the meat goat industry. To better understand the genomic regions and biological pathways contributing to body conformation traits of meat goats, a genome-wide association study was performed using Dazu black goats (DBGs), a Chinese indigenous goat breed. In particular, 150 DBGs were genotyped by whole-genome sequencing, and six body conformation traits, including body height (BH), body length (BL), cannon circumference (CC), chest depth (CD), chest width (CW), and heart girth (HG), were examined. In total, 53 potential SNPs were associated with these body conformation traits. A bioinformatics analysis was performed to evaluate the genes located close to the significant SNPs. Finally, 42 candidate genes (e.g., *PSTPIP2*, *C7orf57*, *CCL19*, *FGF9*, *SGCG*, *FIGN*, and *SIPA1L*) were identified as components of the genetic architecture underlying body conformation traits. Our results provide useful biological information for the improvement of growth performance and have practical applications for genomic selection in goats.

## 1. Introduction

As one of the oldest domesticated livestock, goats provide abundant meat, milk, and fiber [1]. Most of the world’s goat population is found in South-East Asia and Africa, where goats are the major source of meat production [2]. Owing to the rapid growth of the world’s population and increased demand for diversified meat, the goat meat industry has become increasingly important [3]. The history of human migration shaped the genetic structure of goats. One-third of indigenous goat breeds with unique biological characteristics live in Southwest China [4]. Today, there are nearly 120,000 Dazu black goats (DBGs), which are raised in Dazu County of Chongqing (267–934 m above sea level, Southwest China) and are mostly used for meat production [5]. Therefore, the development of candidate genetic markers and functional genes for the optimization of breeding has important practical implications for the goat meat industry.

Growth performance is an economic trait in the meat goat industry. Body conformation traits of goats directly reflect the body size, structure, and development, which are closely related to physiological function and production performance. Body height (BH), body length (BL), heart girth (HG), chest width (CW), chest depth (CD), and cannon circumference (CC) are the most frequently used body conformation traits [6]. However, these traits are generally polygenic and are influenced by environmental factors, making association mapping difficult [7]. Recently, bioinformatics approaches have been applied to study the genetic basis of body conformation traits. The *PRDM6* gene influences growth-related traits, such as CC, CD, and CW, in the early growth and development of goats [8]. *PITX2* is significantly associated with body height and body length in Guanzhong dairy goats and Hainan black goats [9]. Two indels in *PRLR* are significantly correlated with BL, BH, CD, HG, and CC [10].

Genome-wide association studies (GWAS) are a popular approach to detect genome-wide single-nucleotide polymorphisms (SNPs) related to phenotypic traits [11]. To date, GWAS have been widely applied to the goat litter size [12,13], horn status [14], coat color [15], and milk quality [16]. In a study of 4840 New Zealand dairy goats, 43 significant SNPs were associated with fat, proteins, and somatic cell scores [17]. Wang et al. used whole-genome resequencing (WGS) to detect 12 candidate genes from two groups of goats with different litter sizes [18]. Compared to low-density SNP chips, WGS can improve the accuracy and power of GWAS for the detection of SNPs associated with complex traits [19,20]. As the cost of WGS is rapidly decreasing, GWAS using genotypes from WGS have been reported in cattle [21], pigs [22], and goats [23]. In this study, we performed WGS of 150 DBGs and conducted GWAS of six body conformation traits to identify the significant SNPs and related candidate genes.

## 2. Materials and Methods

### 2.1. Ethics Statement

The Chongqing Key Laboratory of Forage & Herbivore approved this experiment. All goat experiments followed the Southwest University Institutional Animal Care and Use Committee regulations (2019, No. GB14925-2010), and no animals were anesthetized or euthanized during the study.

### 2.2. Animals, Phenotypes, and DNA Extraction

A total of 150 female DBGs weighing 35–40 kg from Tengda Farm (Chongqing, China) were used in this study. According to the China National Standard (NY/T 1236-2006) (Appendix A), our team had a specific person to keep the goat calm, a specific person to measure its body conformation traits, another specific person to record the data, and others to calm the other goats. Then, six body conformation traits, (i.e., BH, BL, CC, CD, CW, and HG) of 150 goats were determined (Appendix A). The pedigree information for each goat could be traced back at least three generations. We ensured that there was no direct kinship between these goats. Five milliliters of blood were collected from every individual and brought back to the laboratory using an ice box. In the laboratory, these samples were stored at −80 °C. Next, the standard phenol–chloroform protocol was used to extract the DNA from all blood samples.

### 2.3. Whole-Genome Sequencing

The integrity and concentration of the genomic DNA were assessed by agarose gel electrophoresis and a NanoDrop spectrophotometer (Thermo Fisher Scientific, Waltham, MA, USA), respectively. Then, the sequencing of each sample was performed on the DNBSEQ-T7 platform (Complete Genomics and MGI Tech, Shenzhen, China). According to the standard protocol, the DNA samples were tested, amplified, and purified, and libraries were constructed. During library construction, the genomic DNA was randomly broken into fragments of about 350 bp by the fragmentation instrument. After end repair, the sequencing connectors were added for sequencing. Since the raw data contained reads with junctions or low quality, filtering was performed as follows: (1) reads with adapters were removed, (2) paired reads were removed when the N content exceeded 1% of bases in a read, and (3) paired reads were removed when the quality score ≤ 5.

### 2.4. Quality Control

Clean data were mapped to the goat genome (ARS1) [24] using Burrows–Wheeler Aligner (BWA) (version 0.7.15) [25]. Then, Picard (v 1.129, http://broadinstitute.github.io/picard, accessed on 24 February 2015) and Samtools (v 1.10, https://github.com/samtools/samtools, accessed on 7 December 2019) [25] were used to sort the BAM files. SNPs were identified for all samples using the GATK Unified Genotyper (version 3.4.46) [26]. The detection process was as follows: (1) comparison with the reference genome and de-duplication; (2) the generation of SNP and indel variant files; (3) filtering of the SNPs and indels based on Quality by Depth (QD) < 2.0, root mean square of Mapping Quality (MQ) < 40.0, Fisher Strand (FS) > 60.0, HaplotypeScore > 13.0, and MQRankSum ≤ 12.5; and (4) filtering SNP clusters (i.e., multiple SNPs within 5 bp), SNPs near indels (i.e., within 5 bp), adjacent indels (i.e., indels separated by <10 bp), and loci with GQ (Genotype Quality) less than 20. Then, the following quality control criteria were implemented for the variants detected and individuals: (1) >10% deletions, (2) minor allele frequency (MAF) < 1%, and (3) Hardy–Weinberg equilibrium (HWE) testing with *p* < 1 × 10^−6^. After quality control, all individuals and 16,776,605 SNPs were retained for subsequent analyses.

### 2.5. Population Structure Analysis

A principal components analysis (PCA) was applied to estimate the population structure of 150 goats using the FactoMineR package in R (v4.0.4). The top PCs explaining the largest proportion of variance were extracted to draw scatter plots and to evaluate the population structure [27]. Heat maps were plotted against the kinship matrix to visualize the level of relatedness among individuals within the goat population.

### 2.6. Genome-Wide Association Studies

Since GWAS does not allow missing or unknown SNPs, we used Beagle 5.0 [28] to infer missing data. The mixed linear model used in this study can effectively correct for population structure and complex kinship relationships within the population [29]:*y* = **Xβ** + **Z_k_γ_k_** + *ξ* + *e*
where *y* is a phenotype, **Xβ** is the population structure effect and fixed effects, such as year and season, **Z_k_γ_k_** is the marker effect to be tested; *ξ*~*N* (0, *Kф*^2^) is the polygenic effect, *e*~*N* (0, *Iσ*^2^) is the residual effect, and K is the kinship matrix inferred from the SNPs. Then, we used GEMMA [30] to obtain SNPs significantly associated with body conformation traits. The Bonferroni-corrected *p*-value threshold was 6.01 × 10^−8^, with a corresponding −log *p*-value of 7.22. Significant SNPs were visualized by Manhattan plots, and *p*-value distributions (expected vs. observed *p*-values on a −log_10_ scale) were visualized by a quantile–quantile (QQ) plot.

### 2.7. Functional Enrichment Analysis

Candidate genes were identified according to their physical positions and functions based on the ARS1 reference genome assembly. Significant SNPs were annotated to their nearest genes using ANNOVAR. Gene Ontology (GO) is an international standardized gene function classification system based on GO terms in three broad categories: molecular function (MF), cellular component (CC), and biological process (BP). Kyoto Encyclopedia of Genes and Genomes (KEGG) is the main public database to identify pathways that are significantly enriched for candidate genes compared to reference genes. Gene Ontology (GO) and KEGG pathway analyses of the candidate genes were performed using the clusterProfiler package in R (v4.0.4) [31].

## 3. Results

### 3.1. Phenotypic Data Analysis

The descriptive statistics for six body conformation traits (BH, BL, CC, CD, CW, and HG) are presented in Table 1. To ensure the reliability of the results for follow-up analyses, we estimated the standard deviations, confidence intervals, and coefficients of variation. The dependent variables (BH, BL, etc.) were approximately normally distributed (Appendix A).

### 3.2. Genotypic Data

After library construction and sequencing with the DNBSEQ-T7 system, 17.83 GB of raw data were produced for each individual. After the filtering processes, sequence data for 150 DBGs were mapped to the reference genome to a mean depth of 6×, with mapping rates of 99.06% on average (Table 2). A total of 25,269,456 SNPs and 2,940,245 small indels were in the whole dataset. There were 17,286,575 transitions and 8,127,462 transversions, leading to a Ti/Tv ratio of 2.13 (Appendix A). The SNP distribution in the goat chromosomes is shown in Figure 1A. After quality control, 16,776,605 SNPs from 150 animals were retained for further analysis.

### 3.3. Population Structure Analysis

The population structure was evaluated based on the top three PCs by 3D plots (Figure 1B). In the PCA, three PCs explained 6.45% of the total variance, including 2.32%, 2.12%, and 2.01%. The results indicated that not all individuals clustered together and suggested there was genetic drift between some DBGs and other breeds of goats. Appendix A shows a kinship heat map. To avoid false positives caused by population stratification, PCs and a kinship matrix were used as covariates in the fixed effects model for an association analysis.

### 3.4. Genome-Wide Association Study

The QQ plot in Figure 2 shows the observed and expected *p*-values from the GWAS. Most SNPs did not deviate from the expected *p*-values, suggesting that the models for GWAS were reasonable. After filtering and obtaining adjusted phenotypes, GWAS was performed on the SNPs. Based on stringent thresholds, we found 53 genome-wide significant SNPs (corrected *p*-value < 5.96 × 10^−8^) for BH, BL, CC, and CD and no significant SNPs for CW and HG (Figure 3). Some candidate genes corresponding to the SNPs were identified. The gene corresponding to the significant SNP associated with BH was *ESCHIG00000017110*, located on chromosome 27. For BL, *PSTPIP2* corresponded to the significant SNP located on chromosome 24. We identified 28 genes corresponding to 40 SNPs linked to CC, mainly located on chromosome 8 (*UBAP1*, *CNTFR*, *RPP25L*, *CCL19*, *CCL21*, *TLR4*, *BRINP1*, and *CDK5RAP2*) and chromosome 12 (*FGF9* and *SGCG*). Additionally, 11 significant SNPs associated with CD were located on chromosomes 4, 20, and 22. The genes nearest to these SNPs were *SUN3*, *C7orf57*, *MANEA*, *ENSCHIG00000020259*, *DOK5*, *CBLN4*, *ENSCHIG00000022146*, *CDH9*, *ENSCHIG00000008494*, *CDKAL1*, *SOX4*, and *EDEM1*. (Appendix A).

### 3.5. GO Enrichment and KEGG Analysis

The candidate genes were significantly enriched for several GO terms and KEGG pathways. The top significant GO terms were “GO: 1903319, positive regulation of protein maturation”, “GO: 0048144, fibroblast proliferation”, and other growth-associated GO terms (Appendix A). The KEGG pathway analysis revealed that these candidate genes were highly enriched in four pathways. These pathways included NF-κ B signaling, viral protein interaction with cytokine and cytokine receptors, cytokine–cytokine receptor interactions, and protein processing in the endoplasmic reticulum (Appendix A).

## 4. Discussion

Although thousands of candidate genes underlying economically important traits have been identified in domestic animals, little progress has been made in goat genomics. However, since 2014, low-cost gene chips have had a broad impact on the analysis of the genetic structure of traits of economic interest, as well as on the study of goat population structures at a global scale [32,33]. Luigi-Sierra et al. [8] used the GoatSNP50 BeadChip (Illumina Inc., San Diego, USA) to genotype 825 Murciano-Granadina goats and performed a GWAS. They found that two significant SNPs were associated with the medial suspensory ligament. Based on GoatSNP50 BeadChip, the horn statuses of Boer goats, cashmere goats, and grassland goats were analyzed, revealing a hornless region with a strong selection signal on chromosome 1 [34]. Martin et al. [15] conducted a GWAS of Saanen dairy goats and identified three significant SNPs associated with coat color.

In 2012, a female Yunnan black goat was used to construct a reference genome for the first time by combining Illumina next-generation sequencing and whole-genome mapping techniques [35]. Whole-gene resequencing has become the most rapid and effective method in detecting variation, selection signals, and candidate genes. In Yunshang black goats, GWAS, analyses of runs of homozygosity, and the detection of signatures of selection revealed candidate genes affecting the litter size. Among these candidate genes, some were involved in ovarian function (*PPP2R5C*, *CDC25A*, *ESR1*, *RPS26*, and *SERPINBs*); seasonal reproduction (*DIO3*, *BTG1*, and *CRYM*); and metabolism (*OSBPL8*, *SLC39A5*, and *SERPINBs*) [36]. In addition, Yang et al. [37] used whole-genome sequencing data from 46 Australian Boer goats to detect SNPs and identify genomic regions associated with muscle development. Finally, 30 candidate genes (e.g., *JAK2*, *KCNQ1*, *PDE5A*, *PDLIM5*, and *TBX5*) directly associated with muscle development were obtained. Wang et al. [38] performed whole-genome resequencing on eight Chinese goat breeds in different regions, with a depth of 9–13×. There were about 10 million SNPs per breed, and *Lhx2*, *FGF9*, *Wnt2*, *MC1R*, and *FGF5* may be related to cashmere production traits. Along with the decreasing cost of next-generation sequencing, the application of WGS is becoming increasingly popular [39]. In the future, WGS has the potential to overtake gene chips, which is the most popular sequencing tool at present.

There have been several genomic studies of DBGs to date; however, these studies are limited by small sample sizes. Genome-wide selection signatures in 31 DBGs from two groups with differences in fecundity revealed that *LRP1B* and *GRID2* showed significantly different patterns of linkage disequilibrium [40]. In addition, E et al. [41] classified CNVs according to the previous results and reported three differentially expressed genes (*LOC108633278*, *PPP1R12A*, and *YIPF4*) between the high-yield and low-yield groups. There was a CNV in the *CBLB* gene identified at the individual level. Guan et al. identified some candidate regions based on allele frequency differences in the DBGs and Inner Mongolia cashmere goats. Hormone activity and signaling pathways such as neurohypophyseal hormone activity and adipocytokine signaling pathway [5] may affect reproductive or production traits. In general, additional data are needed to validate the results of these studies of DBGs.

In the GWAS, *PSTPIP2*, which was located on chromosome 24, was significantly associated with BL. According to Lukens et al. [42] and Yao et al. [43], *PSTPIP2* negatively regulates IL-1β, which plays an important role in osteomyelitis. This gene is highly expressed in synovial cells, suppresses the inflammatory response, and reduces the number of osteoclasts by inhibiting the function of fibroblastic synovial cells. In addition, *PSTPIP2* possesses the ability to control the bone resorption process by regulating the assembly of the podosome [44]. Our GO enrichment analysis also revealed that it is related to the molecular function terms actin binding and cytoskeleton binding (Appendix A).

In the GWAS of CD, the following candidate genes were identified: *C7orf57*, *EDEM1*, *SUN3*, *MANEA*, *CDKAL1*, *CDH9*, *CBLN4*, and *SOX4*. Among these, Chio et al. found an association between *C7orf57* and sporadic amyotrophic lateral sclerosis based on 553 patients [45]. *CDKAL1* was identified as an essential gene for the maintenance of normal mitochondrial morphology and function in adipose tissue [46]. In addition, *CDKAL1* affects the transcription of growth hormone genes [47]. *SOX4* regulates apoptosis in cardiomyocytes and controls embryonic and cardiovascular development [48]. KEGG and GO enrichment analyses indicated that it is involved in molecular functions, including DNA binding and the mitotic cell cycle, as well as biological processes, such as positive protein regulation and DNA damage detection (Appendix A).

We detected many significant SNPs associated with CC, especially on chromosomes 8 and 12. For example, *CCL19* on chromosome 8 promotes ligament ossification [49]. A lack of *UBAP1* leads to neurological disorders via the ESCRT pathway, such as hereditary spastic paraplegia [50,51]. Additionally, *FGF9* and *SGCG*, located on chromosome 12, are noteworthy candidates. *FGF9* maintains the osteogenic progenitor cell population by the activation of Akt signaling [52,53]. In contrast, *SGCG* can cause malnutrition and reduce muscle function [54]. Additionally, *FIGN*, which can affect mammalian development [55], and *SIPA1L1*, which upregulates osteoblast development via miR-617/smad3 [56], are potential genes involved in the regulation of the development of muscle and bone. 

We identified many genetic markers that may affect the growth performance of goats. These significant regions have been reported in other studies, supporting the validity of our results. Information on the genomic regions found in this study can facilitate the identification of candidate genes for body conformation traits. However, we did not detect any significant associations for CW and HG, and these results may be due to imprecise phenotypic determinations or an inadequate sample size. In addition to technical factors, the polygenic basis of the morphological traits makes it difficult to detect genetic factors with small phenotypic effects. Ultimately, these results provide guidance for genomic selection in the DBG population for highly efficient genetic improvement.

## 5. Conclusions

In this study, genomic and phenotypic data for 150 DBGs were collected, and genome-wide association studies were carried out. An analysis revealed that *PSTPIP2*, *C7orf57*, *CCL19*, *FGF9*, *SGCG*, *FIGN*, and *SIPA1L* likely contribute to the regulation of bone and muscle development. This was the first large-scale genome survey of DBG, and the results are expected to contribute to genomic selection. Furthermore, our approach can be used for the discovery of genetic markers in goats.

## Figures and Tables

**Figure 1 animals-12-00548-f001:**
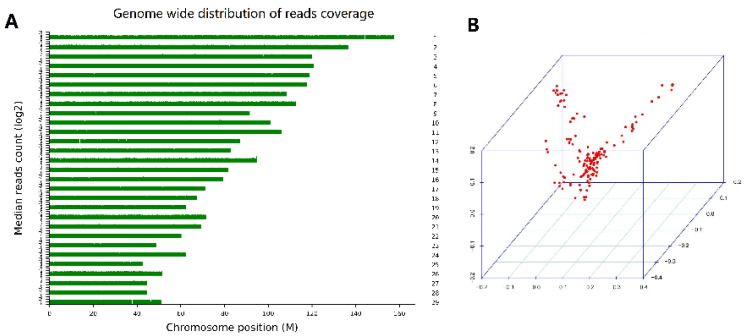
Read density distribution on 29 chromosomes. (**A**) Population structure plots based on 150 goats. The first three principal components (PCs) were used to display the population structure by a 3D plot (**B**).

**Figure 2 animals-12-00548-f002:**
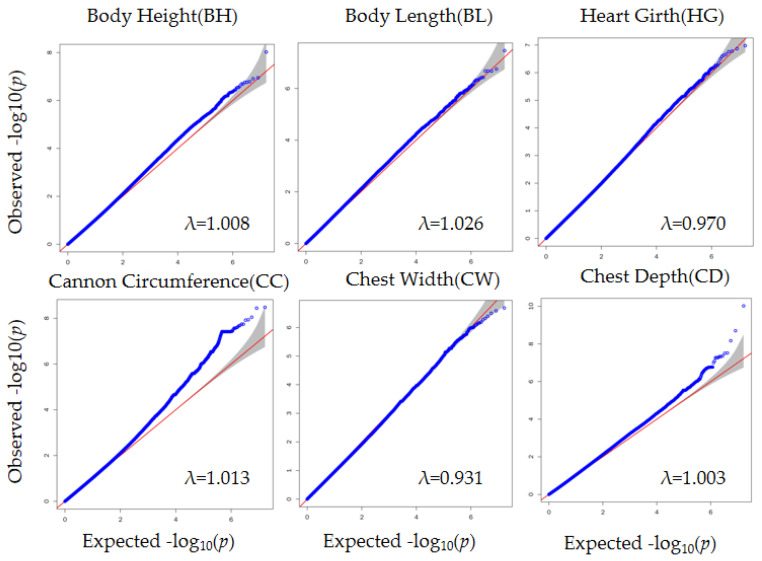
Quantile-quantile (QQ) plots of the six traits (BH, BL, HG, CC, CW, and CD) drawn by the expected *p*-value (the uniformly distributed quantile from 0 to 1) and observed *p*-value for each SNP. The shaded parts are the confidence intervals. λ: genomic inflation factor.

**Figure 3 animals-12-00548-f003:**
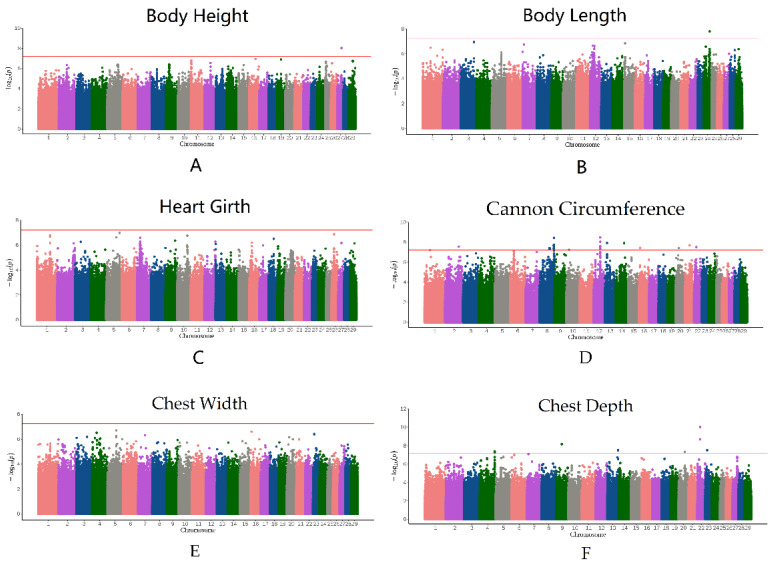
Manhattan plots of the BH (**A**), BL (**B**), HG (**C**), CC (**D**), CW (**E**), and CD (**F**) drawn by the observed *p*-value for each SNP. The red horizontal lines in the Manhattan plots show the significance threshold (6.01 × 10^−8^).

**Table 1 animals-12-00548-t001:** Descriptive statistics of the body conformation traits; *n* = 150.

Trait	Max (cm)	Min (cm)	Mean (cm)	Var	Std. Dev	CV
BH	75.5	50.4	63.52	12.70	3.56	5.6%
BL	78.5	54.3	67.27	79.16	4.38	6.5%
CC	10.3	6.8	8.13	0.33	0.58	7.0%
CD	38.3	20.5	29.78	6.06	2.46	8.2%
CW	23.9	8.2	17.13	4.68	2.16	12.0%
HG	99.5	68.7	81.65	29.60	5.44	6.0%

BH: Body Height; BL: Body Length; CC: Cannon Circumference; CD: Chest depth; CW: Chest Width; HG: Heart Girth; Var: variance; Std.dev: Standard Deviation; CV: Coefficient of Variation.

**Table 2 animals-12-00548-t002:** Summary of the whole-genome sequencing data.

Goat (n)	Raw Bases (Gb) ^1^	Reads Mapped to the Reference ^1^	Total Reads ^1^	Mapping Ratio (%) ^1^	Coverage Depth ^1^
150	17.83	116 293 649	117 573 225	99.06	6.00

^1^ Average per individual.

## Data Availability

The data presented in this study are available on request from the corresponding author.

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
