# Peer review of "Genome-Wide Association Study of Body Conformation Traits by Whole Genome Sequencing in Dazu Black Goats"

_animals, 2022, doi:10.3390/ani12050548_

Round 1

Reviewer 1 Report

The Authors performed whole-genome resequencing of 150 Dazu black goats a Chinese indigenous goat breed and conducted a GWAS analyses of six body conformation traits including body height, body length, cannon circumference, chest depth, chest width, and heart girth, to identify significant SNPs and related candidate genes.

Here are some suggestions that the Authors can consider improving the manuscript.

Introduction

Line 74-76: this sentence anticipate the conclusion. Please move it to conclusion section.

Results

Line 151: please change “ap-proximately” with “approximately”.

Line 156: please change “….depth of 6×” with “….depth of 6X”

Line 168 to 171: In my opinion, the PCA results have been wrongly interpreted. The figure shows three clusters in PC1 (one major and two minors) and two in PC2 and PC3 (the major cluster of PC1 plus one of the two minor of PC1). What are the differences from these three clusters? Are the animals related? Normally the PCA analyses is a dimensionality reduction method that transform a large set of variables into a smaller one that still contains most of the information in the large set. The analyses do not provide information on domestication of animals. Why the Authors define the goat population “……not highly domesticated.”?

Line 197: please add all the name of the genes.

Figure 1: the figure C is too small please enlarge it.

Figure 2: please, add in the figure and in the legend the abbreviations for body conformation traits.

Supplementary Tables and Figure: please add a brief legend explaining the tables and figure.

Author Response

Point 1: Line 74-76: this sentence anticipate the conclusion. Please move it to conclusion section.

Dear reviewer, I appreciate your comments. The reason I wrote this is that in my opinion this paragraph is an introduction to the subsequent experimental section and briefly describes the experimental method and the purpose of the experiment, rather than a conclusion to the experiment as a whole. Now, I have deleted this section.

Point 2: Line 151: please change “ap-proximately” with “approximately”.

Response 2: Sorry for the spelling error, now revised.

Point 3: Line 156: please change “….depth of 6×” with “….depth of 6X”.

Response 3: Revised.

Point 4: Line 168 to 171: In my opinion, the PCA results have been wrongly interpreted. The figure shows three clusters in PC1 (one major and two minors) and two in PC2 and PC3 (the major cluster of PC1 plus one of the two minor of PC1). What are the differences from these three clusters? Are the animals related? Normally the PCA analyses is a dimensionality reduction method that transform a large set of variables into a smaller one that still contains most of the information in the large set. The analyses do not provide information on domestication of animals. Why the Authors define the goat population “……not highly domesticated.”?

Response 4: (1) Dear reviewer, I apologize for the misunderstanding in this section. From the plot of PCA, we find the most goats are clustered together, indicating that only some goats in this population may have crossed with other breeds. We think the reason is that goats were usually bred in the mountains and they were allowed to mate freely in the past.

(2) In the paper, we mention that the Dazu black goats are "not highly domesticated" because we refer to some researches of recent years,i.e.:Comparative genome analyses reveal the unique genetic composition and selection signals underlying the phenotypic characteristics of three Chinese domestic goat breeds(doi: 10.1186/s12711-019-0512-4);The origin of domestication genes in goats(doi: 10.1126/sciadv.aaz5216).They mentioned that the goat in southwestern China is not highly domesticated, and the PCA results in these papers are similar to our study. Therefore, we thought these goats not to be highly domesticated.

(3) “……not highly domesticated.” is not the focus of the article, we have revised the statement in that section to avoid misunderstanding.The revised version is as follows:“ The result indicated that not all individuals clustered together, and suggested there is genetic drift between some DBGs and other breeds of goats.”

Point 5: Line 197: Please add all the names of all genes.

Response 5: Revised.

Point 6: Figure 1: the figure C is too small please enlarge it.

Response 6: This figure has been placed in Figure S2

Point 7: Figure 2: please, add in the figure and in the legend the abbreviations for body conformation traits.

Response 7: Revised.

Point 8: Supplementary Tables and Figure: please add a brief legend explaining the tables and figure.

Response 8: Legend has been added.

Reviewer 2 Report

The manuscript "Genome-wide association study of body conformation traits by whole genome sequencing in Dazu Black Goats" reports putative candidate genes for 4 body measurements based off of a GWAS using SNPs generated from whole genome sequencing rather than a SNP chip. While the methodology is interesting, there are details missing in the methodology, mistakes in the interpretation of the results, and the manuscript would be stronger if there was additional validation beyond presenting the GWAS results.  The manuscript is readable, but there is odd word choice, sentence structure and argument flow throughout that requires additional review.

Specific comments:

Pg1, line 18: were the phenotypes estimated or measured? If estimated, how is this work even relevant?

Pg2, line 86: Who measured the animals? Was it a single person or multiple people. If multiple people, what is the repeatability across the measurers? What is the unit for the measures? This needs to be provided in Table 1 and the supplemental table.

Pg2, line 87: A three generation pedigree is available on the animals, but were single-generation relatives avoided in the sampling, i.e. mother-daughter pairs or full-siblings? If not, why? And if not, did you ensure they were not similar in body measurements?

Pg3, line 110-111: What were the specific filters applied to the SNPs under step 3?

Pg3, line 133: Why is this reference used to describe your Bonferroni cutoff? The reference is not a statistical modeling paper describing the benefits of using or not using a Bonferroni cutoff.

Pg4, line 151: 'approximately' does not have a hyphen; please correct. What statistic was used to determine that the measures follow a normal distribution and where are those results? 

Pg4, line 156 and Table 2: An average of 6x coverage is relatively low quality and surprising for paired-end whole-genome sequencing. How do you know the number of SNPs that were "properly mapped"?

Figure 1, panel c: The legend is too small to read. The figure in general is also too small to interpret. Please fix.

Pg4, line 170-171: How did the authors infer the animals "were not highly domesticated" from the PCA structure when only one breed was used? Did the authors also make comparisons to known non-domesticated goats? If so, that was not mentioned in materials and methods. Based on the figure 1b, there appear to be 3-4 clusters, but why that may be is not addressed by the authors. While the PCs were used in the modeling of the GWAs, the QQ plots provided still display some sort of population structure or bias, especially for cannon circumference, body height and chest depth.

Pg6, line 187: how were the phenotypes adjusted? This wasn't included at all in M&M.

Author Response

Point 1: Pg1, line 18: were the phenotypes estimated or measured? If estimated, how is this work even relevant?

Response 1: The phenotypes in this study were measured by our professional team, and the measurement methods and standards were based on technical specifications for  sheep and goat stud productivity testing (NY/T1236-2006) published by the Chinese Ministry of Agriculture in 2006.

Point 2: Pg2, line 86: Who measured the animals? Was it a single person or multiple people. If multiple people, what is the repeatability across the measurers? What is the unit for the measures? This needs to be provided in Table 1 and the supplemental table.

Response 2: We organized a special team to take care of the measurement of goats in this study. Since goats are large ruminants, measurements cannot be done by one or two persons alone. Therefore, our team had a specific person to keep the goat calm, a specific person to measure its body size, another specific person to record the data, and others to calm the other goats. All the operations can ensure that the experimental data will not be disturbed by human manipulation. In addition, we have added measurement units.

Point 3: Pg2, line 87: A three generation pedigree is available on the animals, but were single-generation relatives avoided in the sampling, i.e. mother-daughter pairs or full-siblings? If not, why? And if not, did you ensure they were not similar in body measurements?

Response 3: In this study, the pedigree information for each goat could be traced back at least three generations. We ensured that there is no direct kinship between these goats. However, it is possible that their ancestors are directly related. Therefore we tested heat map was plotted against the kinship matrix to visualize the level of relatedness among individuals within the goat population. To avoid interference, the mixed linear model (y = Xβ+ Zkγk +ξ+ e) was used in the follow-up GWAS. And k is the kinship matrix inferred from the SNPs.

Point 4: Pg3, line 110-111: What were the specific filters applied to the SNPs under step 3?

Response 4: The SNP filtering criteria used in this study were officially provided by GATK.Refer link:https://sites.google.com/a/broadinstitute.org/legacy-gatk-forum-discussions/tutorials/2806-how-to-apply-hard-filters-to- a-call-set.

QualByDepth(QD): This is the variant confidence (from the QUAL field) divided by the unfiltered depth of non-reference samples.

FisherStrand (FS): Phred-scaled p-value using Fisher’s Exact Test to detect strand bias (the variation being seen on only the forward or only the reverse strand) in the reads. More bias is indicative of false positive calls.

MappingQualityRankSumTest (MQRankSum): This is the u-based z-approximation from the Mann-Whitney Rank Sum Test for mapping qualities (reads with ref bases vs. those with the alternate allele). Note that the mapping quality rank sum test can not be calculated for sites without a mixture of reads showing both the reference and alternate alleles, i.e. this will only be applied to heterozygous calls.

ReadPosRankSumTest (ReadPosRankSum) : This is the u-based z-approximation from the Mann-Whitney Rank Sum Test for the distance from the end of the read for reads with the alternate allele. If the alternate allele is only seen near the ends of reads, this is indicative of error. Note that the read position rank sum test can not be calculated for sites without a mixture of reads showing both the reference and alternate alleles, i.e. this will only be applied to heterozygous calls.

StrandOddsRatio (SOR) 3.0 The StrandOddsRatio annotation is one of several methods that aims to evaluate whether there is strand bias in the data. Higher values indicate more strand bias.

Point 5: Pg3, line 133: Why is this reference used to describe your Bonferroni cutoff? The reference is not a statistical modeling paper describing the benefits of using or not using a Bonferroni cutoff.

Response 5: Dear reviewer, nowadays most of the GWAS performed on animals use SNP chips, and these studies set high Bonferroni thresholds due to the small number of SNP loci that can be obtained from the chips. In this study, genomic data were obtained using whole genome sequencing, and I think that the same criteria cannot be used as for SNP chips. This reference is about GWAS of plant genomic data, which we think can also be used as a reference for our study. Because of cost constraints, GWAS on whole genome data of animals are particularly missing nowadays, and there is also a lack of relevant statistical modeling researches. However, we believe that as the price of next-generation sequencing decreases, there will be more and more genome-wide related studies.Then, this study can also serve as a reference value.

Point 6: Pg4, line 151: 'approximately' does not have a hyphen; please correct. What statistic was used to determine that the measures follow a normal distribution and where are those results? 

Response 6: We apologize for the spelling error here and have revised it. The phenotypic data used in this study were all in Table S1. The analysis was performed by importing the phenotypic data into the software R (v4.0.4) and obtaining frequency histograms and QQ plots for normality detection. Previously, we considered that Figure S1 was sufficient to indicate that the data were in a normal distribution. Thanks to the reviewers' comments, we now add the QQ plots to Figure S1.

Point 7: Pg4, line 156 and Table 2: An average of 6x coverage is relatively low quality and surprising for paired-end whole-genome sequencing. How do you know the number of SNPs that were "properly mapped"?

Response 7: In this study, we performed GWAS on the body type of goats, and only the SNP loci were required. Generally, 5X coverage of whole genome sequencing data is sufficient. Considering the cost of sequencing and sample size, we chose to use 6X coverage. The number of SNPs that were "properly mapped", was obtained by comparing the sequencing results with the reference genome ASR1. Because we only used SNP loci of the results, we needed to protect the data before publication. The raw data will be made public in NCBI after the article is published.

Point 8: Figure 1, panel c: The legend is too small to read. The figure in general is also too small to interpret. Please fix.

Response 8: This figure has been placed in Figure S2.

Point 9: Pg4, line 170-171: How did the authors infer the animals "were not highly domesticated" from the PCA structure when only one breed was used? Did the authors also make comparisons to known non-domesticated goats? If so, that was not mentioned in materials and methods. Based on the figure 1b, there appear to be 3-4 clusters, but why that may be is not addressed by the authors. While the PCs were used in the modeling of the GWAs, the QQ plots provided still display some sort of population structure or bias, especially for cannon circumference, body height and chest depth.

Response 9: 

(1) Dear reviewer, I apologize for the misunderstanding in this section. From the plot of PCA, we find the most goats are clustered together, indicating that only some goats in this population may have crossed with other breeds. We think the reason is that goats were usually bred in the mountains and they were allowed to mate freely in the past.

(2) In the paper, we mention that the Dazu black goats are "not highly domesticated" because we refer to some researches of recent years,i.e.:Comparative genome analyses reveal the unique genetic composition and selection signals underlying the phenotypic characteristics of three Chinese domestic goat breeds(doi: 10.1186/s12711-019-0512-4);The origin of domestication genes in goats(doi: 10.1126/sciadv.aaz5216).They mentioned that the goat in southwestern China is not highly domesticated, and the PCA results in these papers are similar to our study. Therefore, we thought these goats not to be highly domesticated.

(3)We consider the QQ plots in Figure2 as normal because they are drawn by the expected p-value and observed p-value for each SNP. where the significant SNPs will be more deviant than other SNP.We also calculated the λ (genomic inflation factor) of the GWAS results for different phenotypes in order to check the p-value of GWAS. Usually, λ values in the range of 0.95-1.05 indicate that the GWAS results are reliable.  In our study, only the λ value of chest width was not in the range and no significant SNP was found.

(4) “……not highly domesticated.” is not the focus of the article, we have revised the statement in that section to avoid misunderstanding.The revised version is as follows:“ The result indicated that not all individuals clustered together, and suggested there is genetic drift between some DBGs and other breeds of goats.”

Point 10: Pg6, line 187: how were the phenotypes adjusted? This wasn't included at all in M&M.

Response 10: Dear reviewer, I am sorry that I did not make you understand the correct meaning. The processing operation for the phenotype data is mainly in line 85-86, and the results are located in line 147-152, and the details are in Table S1, Table 1, and Figure S1. In general, our adjustment of the phenotype is to observe whether the data are missing, whether they are normally distributed, and to exclude errors.

Reviewer 3 Report

General comments
The manuscript deals with a topic of great interest, both from a scientific point of view and from a more strictly practical point of view.
The association study between productive traits and markers derived from whole genome sequencing are still few at the moment, particularly in the goat species.
Therefore, I believe the proposed study is an important contribution in the field.

In general, the paper is well structured, very well written, with extreme clarity and accuracy. 
The introduction defines the problem adequately, the goal of the work and the experimental design are consistent.
Also the materials and methods section is well structured and clear. 
The used statistical approach for the analysis is appropriate. Although the number of samples is too low to obtain a robust response, I think that, given the scarcity of studies on the specific topic, the obtained results are to be considered interesting.
Results presentation and their discussion could be improved, also taking into account the generalist nature of the journal.
The conclusions are fully supported by the obtained results.
In my opinion, the manuscript only needs minor corrections; I have only a few comments and suggestions reported below.

Specific comments:
L 16: Growth performance and growth rate were not considered in this study. These are traits that, although correlated with the somatic measures, do not correspond to the traits enclosed in presented GWAS.
L 47: m = slm ?
L 51: See previous comment.
L 55: The bibliographic reference does not seem correct in this case. Bangar et al. 2018 do not take into consideration any of the aforementioned traits, their study concerns body weight at different ages (ie. Growth performance).
L 131: The association was conducted using the K matrix " K  is the kinship matrix inferred from the SNPs", then the Genomic Relationship Matrix G as defined in VanRaden 2008.
However, in lines 87 - 88 you write about the pedigree information on the samples used. So I wonder, what is the use of pedigree information, commonly known as matrix A?
Furthermore, having both information available, I believe that a comparison between matrices A and G (calculated based on about 16 million markers) would be interesting and could greatly improve the manuscript.
L 190: Up to what distance was a significant marker considered associated with a gene?
L 197: nearest = see previous comment.
L 146 – 205: In my opinion, some elements, that could make the study more understandable for readers, are missing in the results section.
In general, the results presented in the supplementary tables should be explained in detail. In particular, is the effect shown in table S3 expressed on the detected phenotypes scale or in percentage of variance? Many significant markers seem in the highest linkage, it would be better to calculate and report this element and explain what it implies.
What does the combination of explained variance and degree of linkage imply?
The MAF is calculated and tabulated but is not commented on in the text. How should the variation between 0.01 and 0.38 be considered?
L 206 – 264: The discussion section does not discuss at all the results obtained and presented in this study. A review of works concerning other species and/or other phenotypes is presented. This is more suitable for an introduction than for a discussion.
My advice is to discuss how linkage, effect and MAF can affect the actual use of this information in a genomic improvement plan.

Author Response

Point 1: L 16: Growth performance and growth rate were not considered in this study. These are traits that, although correlated with the somatic measures, do not correspond to the traits enclosed in presented GWAS.

Response 1: Dear reviewer, I apologize we can't conduct a study on specific growth performance and growth rate. The main reason is that there is not enough data on growth performance and growth rate. In this research, the breed,breeding environment and age of these goats are same. Therefore, we think that body conformation traits can reflect growth performance and growth rate of goats.

Point 2: L 47: m = slm ?

Response 2: Sorry, I don't understand what slm means. The “m” means altitude, I have finished revising in L47.

Point 3: L 51: See previous comment.

Response 3: I am sorry for the misunderstanding again. Because we think that body conformation traits can reflect the growth performance and growth rate of goats. The Dazu black goats used in this study were strictly selected in the same breeding environment and the same age. I think this study is related to the growth performance.

Point 4: L 55: The bibliographic reference does not seem correct in this case. Bangar et al. 2018 do not take into consideration any of the aforementioned traits, their study concerns body weight at different ages (ie. Growth performance).

Response 4: I chose this reference because it would provide relevant data from others to indicate common body types. but as reviewer stated, this article is not focused on that.I apologize for misquoting the reference and have revised it.

Point 5: L 131: The association was conducted using the K matrix " K  is the kinship matrix inferred from the SNPs", then the Genomic Relationship Matrix G as defined in VanRaden 2008. However, in lines 87 - 88 you write about the pedigree information on the samples used. So I wonder, what is the use of pedigree information, commonly known as matrix A? Furthermore, having both information available, I believe that a comparison between matrices A and G (calculated based on about 16 million markers) would be interesting and could greatly improve the manuscript.

Response 5: he results between samples due to relatedness.I appreciate the reviewer's idea of comparing matrices A and G. I also acknowledge that this may yield some useful results. But I don't think it will be helpful in the current study.  In subsequent studies of the goat genome, our team will consider conducting relevant research from this perspective.

Point 6: L 190: Up to what distance was a significant marker considered associated with a gene?

Response 6: Genes for which the SNP is intragenic, otherwise the gene is the nearest gene upstream and downstream of the tested SNPs.

Reference: A genome-wide association study reveals candidate genes for the supernumerary nipple phenotype in sheep (Ovis aries), DOI: 10.1111/age.12575

The specific distances are in the sixth column of the table S3, and the distances are in bp.

Point 7: L 197: nearest = see previous comment.

Response 7: As mentioned above.

Point 8: L 146 – 205: In my opinion, some elements, that could make the study more understandable for readers, are missing in the results section.
In general, the results presented in the supplementary tables should be explained in detail. In particular, is the effect shown in table S3 expressed on the detected phenotypes scale or in percentage of variance? Many significant markers seem in the highest linkage, it would be better to calculate and report this element and explain what it implies.

Response 8: The effect shown in table S3 expressed on the detected phenotypes scale. In GWAS, SNP with small but significant effect values are those that have a direct effect on phenotypic values, but the effect is small. SNP with large but insignificant effects are likely to have strong effects. The SNP with great but insignificant effects may have large effects, but are susceptible to environmental effects or due to some anomalies.

For those SNPs in highest linkage, we have planned to conduct experiments in a follow-up study and expect to validate them in a larger number of populations, and also consider adding more goat breeds for comparison.

Point 9: What does the combination of explained variance and degree of linkage imply?

Response 9: I apologize that I did not understand what the reviewer meant by "the combination of explained variance and degree of linkage". The data provided in this article are for completeness and authenticity, and some data may not be of direct use. If the reviewer wants to know the specific role of a data, please indicate the location in the article.

Point 10: The MAF is calculated and tabulated but is not commented on in the text. How should the variation between 0.01 and 0.38 be considered?
Response 10: Because the content of this article is only an early part of our team's work on goat genomic research.We believe that the role of this article is to provide genomic data of Dazu black goat and the SNP loci uncovered by GWAS. Then identify their candidate genes and possible functions.

MAF is widely used in genome-wide association studies of complex traits. SNPs with MAF >0.05 are generally used as the primary target for study, and smaller MAFs may result in false negative results. To study the association of rare mutations with traits, the loss of statistical power is usually compensated by increasing the sample size. In subsequent studies, we will validate the function of these loci in larger populations as well as in relevant cellular experiments.

Point 11: L 206 – 264: The discussion section does not discuss at all the results obtained and presented in this study. A review of works concerning other species and/or other phenotypes is presented. This is more suitable for an introduction than for a discussion. My advice is to discuss how linkage, effect and MAF can affect the actual use of this information in a genomic improvement plan.

Response 11: I appreciate the reviewer's suggestion and I think our team's research idea is similar to yours. As mentioned above, this GWAS is only an early part of our team's research. The ultimate goal of this research is to use the obtained molecular markers for selection and breeding work in goats, such as genomic selection. However, our research are still far from practical application. We will verify the functions of the markers by different techniques in subsequent experiments. In addition, due to cost constraints, much of the work on the practical use of genomes has not yet yielded convincing results, especially for goats.Therefore, we think that this article is not suitable for a discussion of practical use in a genomic improvement plan.

Round 2

Reviewer 1 Report

The authors addressed most of the reviewer concern and made appropriate changes to the manuscript. I think the questions raised were now satisfactorily answered.

Author Response

I am grateful that the reviewers were satisfied with my revisions.

Reviewer 2 Report

The authors have addressed some of my concerns, but not all and still need to make edits within the manuscript.

Line 18: since you actually measured the goats, you need to change the text from "estimated" to "measured".

Line 83: The National Standard for the body measurements is not freely accessible outside of China nor available in English; please include a supplemental file with the translated description of the measurements. The description of how the measurements were collected (i.e. one person collected the measurements) needs to be included in the manuscript.

Line 85: Avoidance of relatives needs to be included in the manuscript text. Stating you have pedigree information that traces back three generations is not enough information to determine if your data was not biased from the onset.

Line 108-109: The actual cutoff values for each filtering criteria are required in the text. This is standard protocol for animal WGS papers. If the authors are following the GATK "best practices", this needs to be cited in the paper; currently only the tool is cited.

Line 123: Was a phased reference panel used in the imputation? If so, what was the source of the reference panel?

Line 129-130: The GEMMA sentence needs to be re-written. Significant SNPs for what? Which model in GEMMA did you use? 

Line 130-131: The grammar needs to be fixed in this sentence; it needs a verb. The plant paper citation is inappropriate to use and needs to be removed. Other animal WGS papers have applied Bonferroni cutoffs and do not use a citation. It is common knowledge that Bonferroni cutoffs are based on the total number of SNPs in the dataset, and therefore will change based on the dataset.

Line 155: change "in average" to "on average". 

Table 2: "Properly mapped" is not the appropriate terminology. While the authors have explained the meaning of the header in their response, the header still needs to be changed to reflect SNPs mapped to the reference.

Figure 1 (and 2): I appreciate the updated explanation about the PCs. Out of curiosity, did you compare the PC1 scores to the body measurements to ensure the clusters had variation in the body measurements, especially cannon circumference?  I ask because your QQ plot shows an early and wide deviation and a plateau in cannon circumference; while your lambda does fall within acceptable ranges, you still need to consider the shape of the SNP QQ plot and for CC it looks suspicious. 

Author Response

Point 1: Line 18: since you actually measured the goats, you need to change the text from "estimated" to "measured".

Response 1: Sorry for this error, now revised.

Point 2: Line 83: The National Standard for the body measurements is not freely accessible outside of China nor available in English; please include a supplemental file with the translated description of the measurements. The description of how the measurements were collected (i.e. one person collected the measurements) needs to be included in the manuscript.

Response 2: We have provided the measurement standard for goats in supplementary documents (Figure S1: The National Standard for the body measurements of Goats in China).

The specific measurements have been added to the manuscript. 

“According to the China National Standard (NY/T 1236-2006) (Figure S1), our team had a specific person to keep the goat calm, a specific person to measure its body conformation traits, another specific person to record the data, and others to calm the other goats”

Point 3: Line 85: Avoidance of relatives needs to be included in the manuscript text. Stating you have pedigree information that traces back three generations is not enough information to determine if your data was not biased from the onset.

Response 3: Revised. 

“We ensured that there is no direct kinship between these goats”

Point 4: Line 108-109: The actual cutoff values for each filtering criteria are required in the text. This is standard protocol for animal WGS papers. If the authors are following the GATK "best practices", this needs to be cited in the paper; currently only the tool is cited.

Response 4: We have added the actual cutoff values. 

“Quality by Depth (QD) < 2.0, root mean square of Mapping Quality (MQ) < 40.0, Fisher Strand (FS) > 60.0, HaplotypeScore >13.0, and MQRankSum ≤ 12.5”

Point 5: Line 123: Was a phased reference panel used in the imputation? If so, what was the source of the reference panel?

Response 5: (1) We chose ARS1 as the reference panel. Because this assembly represents a similar to 400-fold improvement in continuity due to properly assembled gaps, compared to the previously published C. hircus assembly, and better resolves repetitive structures longer than 1 kb, representing the largest repeat family and immune gene complex yet produced for an individual of a ruminant species. The reference panel plays the primary role in determining the accuracy of imputed variants. Imputation accuracy for a variant generally increases with increasing reference panel size, and variants must be present in the reference panel in order to be accurately imputed.

(2) Genotype imputation is based on identity by descent (IBD). Two chromosome segments that are inherited from a common ancestor without recombination since the common ancestor are said to be inherited identical by descent. In an IBD segment, the two chromosomes will have identical allele sequences except at sites that have mutated in one of the lineages since the common ancestor. We can use the genotypes at the target markers to identify long IBD segments that a target haplotype shares with the reference haplotypes. If an IBD segment is accurately identified, the ungenotyped alleles in the IBD segment in the target haplotype can be copied from the IBD segment in the reference haplotype. Since there is uncertainty in inferring IBD, a probabilistic model is used to account for the uncertainty and to produce a posterior probability for each possible allele at an imputed marker on the target haplotype. This probabilistic model is typically a hidden Markov model.

Point 6: Line 129-130: The GEMMA sentence needs to be re-written. Significant SNPs for what? Which model in GEMMA did you use? 

Response 6: I'm sorry the reviewers didn't understand this paragraph, and I've now revised. 

“Then we used GEMMA to obtain some SNPs significantly associated with body conformation traits”

(1) GEMMA is a method that can solve mixed linear models quickly and accurately.

(2) The model used in GEMMA is the above mentioned y = Xβ + Zkγk + ξ + e. We do not think it is necessary to repeat the description here.

Point 7: Line 130-131: The grammar needs to be fixed in this sentence; it needs a verb. The plant paper citation is inappropriate to use and needs to be removed. Other animal WGS papers have applied Bonferroni cutoffs and do not use a citation. It is common knowledge that Bonferroni cutoffs are based on the total number of SNPs in the dataset, and therefore will change based on the dataset.

Response 7: Revised. “The Bonferroni-corrected p-value threshold was 6.01 × 10–8, with a corresponding −log p-value of 7.22. ”

Point 8: Line 155: change "in average" to "on average". 

Response 8: Revised.

Point 9: Table 2: "Properly mapped" is not the appropriate terminology. While the authors have explained the meaning of the header in their response, the header still needs to be changed to reflect SNPs mapped to the reference.

Response 9: Revised.

“Reads mapped to the reference”

Point 10: Figure 1 (and 2): I appreciate the updated explanation about the PCs. Out of curiosity, did you compare the PC1 scores to the body measurements to ensure the clusters had variation in the body measurements, especially cannon circumference?  I ask because your QQ plot shows an early and wide deviation and a plateau in cannon circumference; while your lambda does fall within acceptable ranges, you still need to consider the shape of the SNP QQ plot and for CC it looks suspicious. 

Response 10: We are grateful to the reviewer for the approval. Before the experiment, we did not know whether there were differences between body measurements of goats. Therefore, we chose six different body size indicators to measure. Through this study, we obtianed some SNPs and candidate genes. We can understand the reviewer's doubts, and we cannot guarantee that the results are percentage correct. In order to explore whether the results are correct enough, we plan to validate these SNPs in larger and more populations in our subsequent experimental program and consider using cellular experiments to validate the function as well.

In this experiment, our results show that there is a difference between clusters between some body measurements. Regarding the QQ chart, we believe that the deviations in the chart are within a reasonable range. Firstly lambda is within the acceptable range. Secondly there are similar shapes in QQ plots in other literature. We also thank the reviewers for their comments, and in the follow-up experiments we will focus on whether clusters differ between body measurements.

Refer link:

https://doi.org/10.1007/s13258-020-00937-5

https://doi.org/10.3390/ani11071927
